# Acute and Rapid Response of *Melissa officinalis* and *Mentha spicata* to Saline Reclaimed Water in Terms of Water Relations, Hormones, Amino Acids and Plant Oxylipins

**DOI:** 10.3390/plants11243427

**Published:** 2022-12-08

**Authors:** María José Gómez-Bellot, Beatriz Lorente, Sonia Medina, Ángel Gil-Izquierdo, Thierry Durand, Jean-Marie Galano, Sergio Vicente-Sánchez, María Fernanda Ortuño, María Jesús Sánchez-Blanco

**Affiliations:** 1Department of Irrigation, CEBAS-CSIC, University Campus of Espinardo–Edif. 25, 30100 Espinardo, Spain; 2Research Group on Quality, Safety and Bioactivity of Plant Foods, Department of Food Science and Technology, CEBAS-CSIC, University Campus of Espinardo–Edif. 25, 30100 Espinardo, Spain; 3Institut des Biomolécules Max Mousseron (IBMM), Pôle Chimie Balard Recherche, UMR 5247, CNRS, University of Montpellier, ENSCM, 34090 Montpellier, France; 4Murcia Health Service, C/Central nº 7. Edif. Habitamia, 30100 Murcia, Spain

**Keywords:** salinity, aromatic plants, leaf water potential, gas exchange, hormonal response, phytoprostanes

## Abstract

The use of reclaimed water is considered an efficient tool for agricultural irrigation; however, the high salinity associated to this water could compromise plant quality and yields. Balm and spearmint plants were submitted for 15 days to three irrigation treatments in a controlled chamber: control with EC: 1.2 dS m^−1^ (control), reclaimed water from secondary effluent (EC: 1.6 dS m^−1^) (S) and water from secondary effluent with brine (EC: 4.4 dS m^−1^) (SB). The plant water status, stomatal and hormonal regulation, nutritional response, concentration of amino acids and plant oxidative stress-based markers, as well as growth were evaluated. Both species irrigated with saline reclaimed water reduced leaf water potential and gas exchange in comparison with control plants, following 2 days of exposure to irrigation treatments. Nevertheless, spearmint plants recovered photosynthetic activity from the seventh day onwards, maintaining growth. This was attributed to hormonal changes and a greater accumulation of some amino acids and some plant oxylipins (phytoprostanes) in comparison to balm plants, which contributed to the improvement in the organoleptic and health-promoting properties of spearmint. A longer irrigation period with saline reclaimed water would be necessary to assess whether the quality of both species, especially spearmint, could further improve without compromising their growth.

## 1. Introduction

In the near future, it is believed that water scarcity will become the biggest problem and the most complicated challenge faced by agriculture, especially in Mediterranean areas [1,2]. Therefore, there is a strong need to search for new crop practices and technologies, including deficit irrigation application, high-tech irrigation systems, less-water-demanding crops, water reuse and low-quality water for crop irrigation needs [3]. In this sense, the use of reclaimed wastewater, brine water or its desalination using reverse osmosis is considered an efficient tool for managing water resources [4,5,6] since it may increase agricultural production in regions that cope with irregular and scarce water availability, contributing to food safety [7]. This kind of water may be a potential source of nutrients, with the consequent reduction in fertilizers. However, the reuse of wastewater for agriculture may have some risks and limitations, derived precisely from its chemical composition, which will determine its suitability for irrigation. A very common problem associated with the reclaimed wastewater is the high concentration of saline ions, such as Cl and Na, which may accumulate in crops over time, compromising plant quality and yields [8]. In coastal regions, treatment plants located near the sea often have higher salinity than those inland. In fact, desalination of sea water and brackish water is a common technology in these treatment plants. It should be noted that the high salinity of brine derived from seawater desalination has significant environmental effects. Therefore, the proper management of brine is of vital importance to reduce the environmental impact as much as possible, whether it is for reusing brine in sectors such as agriculture or to discharge it back into the sea [9,10].

The physiological responses of a plant to salinity are often complex and multifaceted. In general, high salinity in irrigation water induces the absorption of toxic ions, nutrient imbalance, difficulties taking water from the soil, photosynthesis inhibition and metabolic changes [11]. The first plant response to salinity is the alteration of water relations due to an osmotic stress, which occurs within minutes to days after the application of salinity conditions. As a result, the plant closes its stomata and reduces its photosynthetic capacity, and, consequently, plant growth is reduced. The second plant response to salinity takes place over a longer period (days to weeks) and involves the build-up of ions in the shoot to toxic concentrations, causing ion toxicity, ion imbalance and the inhibition of enzyme activity. As a result, there is a premature senescence of leaves and, ultimately, reduced yield [12]. The extent of the adverse impact of salinity on the previously mentioned physiological processes depends on the salinity levels, exposure period [13] and species. It is obvious from the considerations above that a plant has multiple options to respond to salt. The ideal alternatives would be avoiding the take up of salt and achieving osmotic adjustment through the accumulation of other solutes in cells, or using Na and Cl as osmolytes in the vacuole of cells while maintaining cytosolic levels of these ions in a non-toxic range and accumulating solutes that are compatible with cell functions in the cytoplasm. However, it is not clear which mechanisms plants employ to maintain residual growth and to what extent these mechanisms differ between short- and long-term responses [14]. Improving salinity stress tolerance in crops is one of the current major objectives in achieving sustainable agriculture. To achieve this goal, it is necessary to understand how salinity affects the morphological, physiological and biochemical properties of plants [15].

Aromatic and medicinal plants are cultivated for different plant parts and their active constituents are used in several fields such as agrifood, perfumes, pharmaceutical industries and natural cosmetic products [16]. Aromatic plants are valued for their aroma and flavor, although most of them have, at the same time, medicinal properties thanks to their richness in bioactive compounds, which serve as active ingredients in the manufacture of many medicines [17]. Moreover, the biosynthesis of their constituents, mainly secondary metabolites, is strongly influenced by environmental factors, such as salinity [18]. A wide range of research has studied the effect of salinity in the physiological behavior of these plants, with different findings [19,20,21]. The vast majority of these works have focused on phenolic compounds, essential oil yield, and hormonal and antioxidant activity. Recently, plant oxylipins have emerged as new types of secondary metabolites beyond polyphenols or other classic plant constituents [22]. These compounds are generated from the oxidation of linolenic acid present in cell membranes caused by reactive oxygen species (ROS) giving place to a wide range of phytoprostanes including the prostaglandin D1, E1, F1, A1, B1, or the deoxy J1-ring system, as well as malondialdehyde [22,23]. Regarding aromatic plants, no attention has been paid to the dynamics of these compounds and amino acids (precursor compounds of several hormones), which are considered excellent compounds not only as defense mechanisms against abiotic stresses but also to contribute to the quality of the product, in terms of organoleptic characteristics and composition [24]. In fact, several phytoprostanes are intermediate compounds in the jasmonic acid synthesis being catalyzed by lipoxygenases where linolenic acid is converted to 12, 13- (S) -epoxy-octadecanoic acid (12, 13-EOT) [25]. Therefore, the participation of these compounds in the redox balance of the plant has been suggested in different crops [26,27]. In this sense, the use of saline wastewater in plants could provide significant beneficial characteristics to aromatic and medicinal plants since qualitative and quantitative variation in plant oxylipins could provide bioactivities to human health because of their structural and functional matches with human oxylipins [28,29]. Several authors have shown that irrigation with reclaimed wastewater is possible in lemon grass, basil, lemon verbena and oregano without a reduction in growth and yield [30,31], although in those studies the irrigation water was not saline.

Based on the above, we evaluate: (1) whether the irrigation with different saline reclaimed wastewater is suitable to induce different changes in *Melissa officinalis* and *Mentha spicata* under controlled conditions during the short term, concerning the physiological aspects such as plant water relations and gas exchange and the metabolism process linked to hormones, amino acids and oxylipins; (2) whether the characteristics of this water may stimulate the biosynthesis of promising health-promoting compounds in these plants, improving their quality.

## 2. Results

### 2.1. Relative Chlorophyll Content, Water Relations and Gas Exchange

Both species showed a reduction in the relative chlorophyll content when they were irrigated with SB water. This reduction was observed from the 7th day in the case of balm plants, and at the 15th day in the case of spearmint plants (Figure 1).

Both species irrigated with S and SB water showed a reduction in Ψleaf in comparison with control plants at the 2nd day (Figure 2A,D). Curiously, spearmint plants irrigated with SB treatment recovered these values after 7 days, which were close to the control plants at the 15th day (Figure 2D). Balm irrigated with SB treatment showed a decrease in Ψs in comparison with those irrigated with control water, while spearmint irrigated with SB treatment showed it only at the 15th day (Figure 2B,E). Results from Ψ100s showed that osmotic adjustment was not observed in either species in a short-period (Figure 2C,F).

Balm plants irrigated with S treatment were able to maintain gas exchange levels close to those irrigated with control water at the 2nd day. Balm plants irrigated with SB treatment reduced both their stomatal conductance and photosynthetic rate from the 7th day (Figure 3A,C). In line with leaf water potential results, spearmint irrigated with S and SB treatments showed a reduction in gas exchange only at the 2nd day of the experiment, while there were no significant differences between treatments from the seventh day onwards (Figure 3B,D).

### 2.2. Leaf Mineral Content and Plant Growth

Leaf mineral content in balm plants was not statistically affected by the type of water used at the 15th day, except by a reduction in P ion in leaves after the irrigation of SB treatment (Table 1). Conversely, spearmint plants irrigated with SB treatment showed an accumulation of Cl^−^ and Na^+^ in leaves at that time.

Balm plants reduced leaf number as salinity increased in the irrigation water. A reduction in leaf area as well as leaf and root dry weight was also observed in these plants irrigated with SB treatment (Table 2). There were no symptoms of ion toxicity but there was a quick wilting of leaves due to the lack of water. In spearmint plants, growth parameters were hardly affected by S treatment, except for an increase in leaf dry weight (DW). Spearmint plants irrigated with SB treatment showed a reduction in leaf area and an increase in root dry weight. No symptoms of ionic toxicity were observed either (Table 2).

### 2.3. Concentration of Phytohormones, Amino Acids and Phytoprostanes in Leaves

Several phytohormones were identified in leaves in both species at the 15th day after imposing treatments: the precursor of ethylene, 1-aminocyclopropane-1-carboxylic acid (ACC), cytokinins (TZ, TZ-rib and TZ-glc), abscisic acid (ABA), salicylic acid (SA) and the phytoalexin scopoletin (SC) (Table 3). A reduction in ABA and SA content was observed in the leaves of balm plants irrigated with S treatment, while the irrigation with SB treatment accumulated ACC, TZ and TZ rib and reduced TZ-glc content in the same plants. An increase in SC content was observed in spearmint plants irrigated with the S treatment, while an accumulation of ACC and ABA content, as well as a reduction in TZ and TZ-glc was observed in the leaves of spearmint plants irrigated with both saline treatments (Table 3).

As regards amino acids, some of them were modified by the type of irrigation water at the 15th day (Table 4). A higher concentration of aspartic acid was observed in balm plants irrigated with both saline treatments compared with those irrigated with the control treatment. A higher concentration of serine and methylhistidine was observed in spearmint plants irrigated with SB treatment, and a higher concentration of hydroxyproline and leucine was observed in spearmint plants irrigated with both saline treatments compared with those irrigated with the control treatment (Table 4).

The total phytoprostanes retrieved from the analyses indicated that their concentrations were not significant, regardless of the irrigation treatment in balm, although a slight augmentation was tested under SB conditions. The total phytoprostanes were increased in spearmint under S conditions and decreased after SB irrigation at the 15th day (Table 5).

The total phytofurans were present at much higher concentrations than phytoprostanes (Table 5). These compounds decreased in a significant manner in balm and spearmint under the S and SB irrigation treatments compared to the control (Table 5).

Regarding the eight individual phytoprostanes analyzed, only four of these plant oxylipins were detected in both types of plants (Table 5). The concentration of phytoprostanes in balm plants was not statistically affected by the type of water at the 15th day, although a tendency to increase 9-F1t-PhytoP and 9-epi-9-F1t-PhytoP concentrations by the irrigation of SB treatment was observed (Table 5). A higher concentration of 9-F1t-PhytoP and a slight increase in 9-epi-9-F1t-PhytoP was observed in spearmint plants irrigated with S treatment. In the case of ent-16-epi-16-F1t-PhytoP + ent-16-F1t-PhytoP, these types of oxylipins stayed unchanged regardless of the irrigation treatment.

Concerning the individual phytofurans, all series of the phytofurans analysed were detected in balm, while only two were present in spearmint for the control and SB treatments. The quantitative analysis retrieved in the current work for these compounds provided much higher concentrations of phytofurans than the individual phytoprostanes (Table 5). For balm and spearmint, ent-9(RS)-12-epi-ST-Δ10-13-PhytoF was the most important compound since it decreased in the S and SB irrigation types, while ent-16(RS)-13-epi-ST-Δ14-9-PhytoF only increased under S irrigation conditions in spearmint (Table 5).

## 3. Discussion

Although the use of non-conventional water is an alternative that is spreading in many regions, the irrigation of aromatic and medicinal crops with saline reclaimed water has not been explored. Despite the short time that elapsed after the imposition of treatments (2 days), both species irrigated with saline treatments showed a reduction in leaf water potential as a consequence of a higher water retention [32], causing a reduction in gas exchange at the 2nd day after the application of treatments [33]. All these changes have been reported as the first common response of the plant to salinity [34]. Despite the reduced leaf water potential in balm plants irrigated with S treatment, the net photosynthetic rate and stomatal conductance values were similar to control plants during the experiment, indicating that dehydration in balm plants was not enough to cause a reduction in gas exchange, at least when they were irrigated at 1.6 dS m^−1^. Spearmint plants showed a reduction in water relations and gas exchange only at the 2nd day, suggesting that these plants were able to readjust their physiological response to the new conditions, as an intrinsic response of the plant to salinity.

Although there were hardly any changes in the nutritional response of these plants after fifteen days with saline irrigation, clear differences were observed between both species in the absorption mechanism of mineral elements. While balm plants only showed a gradual reduction in P as salinity increased in the irrigation water, spearmint showed an accumulation of Na^+^ and Cl^−^. Nevertheless, toxic levels in spearmint plants were not reached fifteen days after the application of treatments, as the results of biomass parameters (leaf DW) showed. It should be noted that although both reclaimed waters, especially SB treatment, contained higher levels of beneficial ions than the control water, which could contribute as a fertilizer, there was not an accumulation of these elements in the leaves of both plants [35]. Biomass parameters indicated a marked reduction in aerial and root growth in balm plants irrigated with SB treatment. Our results are not in concordance with [36], who reported that the biomass of *Melissa officinalis* was not negatively affected by the irrigation of 50 Mm NaCl (≈4.8 dS m^−1^) during ten days. In our case, there were no symptoms of ionic toxicity in balm plants. Thus, the wilting observed in balm plants irrigated with SB treatment during the last week of the experiment was attributed to the difficulty to absorb water from the substrate and, therefore, to the limitation of stomatal conductance and the reduction in the chlorophyll content [37,38]. On the other hand, spearmint plants were able to maintain aerial biomass by increasing root DW and reducing leaf area with a similar leaf number. These results are in concordance with other studies, which reported that the application of saline water at levels between 20–50 mM NaCl (≈ 1.8–4.8 dS m^−1^) during 25–42 days did not significantly affect the growth and biochemical functions of *Mentha spicata* plants [3,39], while others affirmed that these plants cannot tolerate salinity levels above 30 mm (≈ 2.7 dS m^−1^) [40].

Phytohormones under salinity conditions have been widely studied in many crops, including aromatic and medicinal species [41,42], due to their essential role in salt tolerance. Fifteen days after the application of treatments, hormonal regulation in balm plants was related to plant sensitivity to salinity, while changes observed in spearmint plants were related to conferring tolerance to salinity. Balm irrigated with SB treatment accumulated cytokinins (TZ and TZ-rib) as well as ACC, and those irrigated with S treatment decreased SA and ABA. The reduction in CZ transport to shoots alters the expression of certain genes, causing physiological responses that help to cope with stress [43], while the accumulation of CKs induces a delay of stomata closure, facilitating water losses and magnifying the effect of salinity [44]. These changes could explain the drastic reduction in photosynthesis and, thus, the reduction in growth in balm plants irrigated with SB treatment. Spearmint irrigated with saline treatments activated the biosynthesis of ABA and SC, which have been extensively described as a tolerance mechanism to environmental stresses [45,46]. ABA allows the stomatal closure, thus avoiding water losses, while SC helps to activate useful chemical processes. The precursor of ethylene (ACC) was accumulated by both species, although balm plants to a greater extent than spearmint (Figure 3), which has been linked to a greater sensitivity to salinity by several authors [47,48]. However, the controversial findings of both ethylene and ACC in plants suggest that salinity stress can act on ET production, having a negative or positive role in the regulation of plant stress tolerance [41].

The dynamic in amino acids’ content also differed from balm to spearmint plants irrigated with saline reclaimed water. The accumulation of amino acids in the leaves of spearmint was more evident than in the leaves of balm. A higher level of amino acids has been associated to tolerant species to salinity [49]. They can influence flavor, vitamin content and, thus, in the quality of fruits, and, to a lesser extent, leaves [50]. According to [51], the increase in leucine and met-histidine, which was found in spearmint irrigated with saline reclaimed water, improves the nutritional quality of foods from vegetable origin. On the other hand, the molecular taste receptor T1R1 + 3, found in humans, responds to amino acids such as aspartic acid, which significantly increased in balm plants [52]. It should be noted that proline was not detected in these species. Proline is an important amino acid involved in promoting osmotic adjustment, a mechanism that was not observed in these plants.

Regarding the phytoprostanes and phytofurans, related plant oxylipins would provide supportive information about the influence of the irrigation treatments with different reclaimed water and salinity conditions on the generation of oxidative stress in balm and spearmint plants and the possible useful accumulation of these compounds concerning the improvement in the healthy qualitative properties of these plants [28,29]. The concentration of phytoprostanes in balm and spearmint leaves were in the same range as other plant species such as Agrostis tenuis, rice, nut kernels, some types of almond cultivars and passion fruits [28]. Concerning the mechanistic generation of these compounds in this study, they were produced only by the radical attack of ROS to linolenic acid and not in an enzymatic way related to jasmonic acid since this plant hormone was detected under the limit of quantification. It is noteworthy that the phytofurans were found in much higher concentrations than phytoprostanes in baseline control plants (balm and spearmint). This issue was only previously described in *Cucumis melo* leaves [53] and some brown macroalgae [28] since the phytoprostanes are usually more concentrated in plant species than phytofurans. This is the first study considering plant oxylipins in balm and spearmint, and, therefore, there is no previous information to corroborate if the abundance in the quantity of phytofurans is the proper plant physiology of these plants species or whether this could have been induced by intrinsic external biotic and abiotic stresses present in the cultivation zones. Total phytoprostanes were not affected by the soft and high salinity conditions in the balm species. However, the phytofurans were dramatically affected after S and SB irrigations at the 15th day of the treatments. For both species, these compounds decreased in a significant manner regardless of the irrigation treatment. It is generally assumed that high salinity induces oxidative stress by the increase in the reactive oxygen species (ROS) in some plant species. Salt stress provokes proline and hydroxyproline in plants [54]; hydroxyproline is a toxic analog of proline that promotes proline overaccumulation [55]. Spearmint and balm were able to accumulate hydroxyproline but not proline under S and SB salt stress conditions, the authentic osmotic regulator indicating a deficient metabolism and a possible stomatal opening by its damage caused by sodium ions and an imbalance of the transpiration regulation [56]. The stomata are the plant structure for gas exchange regulation and if they are open in balm and spearmint under average and high salinity irrigation conditions, it leads us to think that the oxygen outlet is increased, decreasing the oxygen accumulation and ROS generation, and producing a downregulation in phytofuran production. This mechanistic effect of phytofuran was contrary to that found in *Cucumis melo* leaves under thermal stress conditions of cultivation where stomatal structures were not injured [53]. As regards the accumulation of individual phytoprostanes, 9-F1t-PhytoP increased in the leaves of spearmint plants irrigated with S treatment. Therefore, and according to the above reasons, spearmint plants irrigated with S treatment kept the stomatal structure by closing its system, accumulating oxygen and ROS generation, and, as a consequence, increasing phytoprostane production. The individual phytofuran ent-9(RS)-12-epi-ST-Δ10-13-PhytoF was the most important compound responsible for the decrease in total phytofurans in spearmint and balm under S and SB irrigation conditions. Therefore, it could be postulated as a good marker of the dysregulation of stomatal function under salt stress conditions.

## 4. Materials and Methods

### 4.1. Plant Material and Experiment Conditions

The experiment was performed on two aromatic species, *Melissa officinalis* (n = 54) and *Mentha spicata* (n = 54), collected from a nursery; each had a height of 10 cm. Plants were transplanted into plastic pots (1.5 L) filled with a commercial soilless substrate composed of peat, coconut fiber and perlite (67/30/3, *v/v/v*) (Fertiberia S.A., Madrid, Spain), and grown in a controlled growth chamber with the following conditions: natural temperature, 24 °C/18 °C (day/night); photosynthetic photon flux density, 350 μmol m^−2^ s^−1^; photoperiod, 16/8 h (light/dark) and relative humidity of 60%. During transplantation, plants were amended with 2 g L^−1^ of Osmocote Plus (14:13:13 N,P,K plus microelements). Plants were watered manually, using tap water whose electrical conductivity was 1.0 dS m-1. Field capacity of substrate was calculated according to [57]. Each pot was weighed before each irrigation event, and the volume of irrigation water required to refill the pot to its threshold level was calculated and added to each plant.

### 4.2. Irrigation Water Treatments and Experimental Design

Fifteen days after transplant and for each species, three irrigation treatments were applied at 100% water holding capacity: control or fresh water (C) (EC: 1.1 dS m^−1^); water from the secondary effluent (S) (EC≈1.6 dS m^−1^), which was reclaimed after feeding wastewater to secondary treatments in a municipal wastewater treatment plant (WWTP) located in Balsicas (Murcia, Spain) (latitude 37° 47′48″ N, longitude 0° 57′36″ W) ; and a mixture of water from the secondary effluent and water residue or brine (SB) (EC≈4.4 dS m^−1^) from the same WWTP. The duration of the experiment was fifteen days from the beginning of the treatments. The experimental plot consisted of three treatments with three replicates per treatment. Therefore, there were eighteen plants per treatment and six plants per replicate. The physicochemical characteristics of the irrigation water from control, S and SB treatments are presented in Table 6. Water analysis from treatment SB showed the highest concentration of most of the components, while water from S treatment showed an intermediate ion concentration between the control and B treatments (Table 6).

### 4.3. Water Relations

Leaf water relations were measured two days, seven days and fifteen days after the application of treatments, in six plants per treatment (two plants per replication). Leaf water potential (Ψleaf) was measured collecting a mature leaf according to [58] using a pressure chamber (Model 3000; Soil Moisture Equipment Co., Santa Barbara, CA, USA). Leaves were placed in the chamber within 20 s of collection and pressurized at a rate of 0.02 MPa s−1. Adjacent leaves were also collected, frozen immediately in liquid nitrogen (−196 °C) and subsequently stored at −30 °C. After thawing, the leaf osmotic potential (Ψos) was measured in the extracted sap using a WESCOR 5520 vapor pressure osmometer (Wescor Inc., Logan, UT, USA), according to [59]. The leaf osmotic potential at full turgor (Ψ100s) was estimated as indicated above for Ψos, after placed in distilled water overnight to reach full saturation.

### 4.4. Gas Exchange and Relative Chlorophyll Content

The leaf photosynthetic rate (Pn) and stomatal conductance (gs) were measured two days, seven days and fifteen days after the application of treatments, in six plants per treatment (two plants per replication), using a gas exchange system (LI-6400; LI-COR Inc., Lincoln, NE, USA). The reference CO_2_, photosynthetically active radiation (PAR) and speed of the circulating air flow inside the system were set at 400 ppm, at 1500 µmol m^−2^ s^−1^ and at 500 µmol s^−1^, respectively.

The relative chlorophyll content (RCC) was determined on the same days as the rest of measurements, in six fully opened leaves per treatment (two plants per replicate) and using a Minolta SPAD-502 chlorophyll meter (Konica Minolta Sensing Inc., Osaka, Japan).

### 4.5. Chemical and Reagents

Analytical standards of the phytohormones 1-aminocyclopropane-1-carboxylic acid, gibberellic-5 acid, trans-Zeatin glucoside, abscisic acid, salicylic acid and scopoletin were purchased from Santa Cruz Biotechnologies (Dallas, TX, USA). Trans-Zeatin, trans-Zeatin riboside and [2H5]-trans-Zeatin were obtained from Olchemlm (Olomouc, Czech Republic). Analytical standards of the PhytoPs: 9-F1t-PhytoP; ent-16-F1t-PhytoP, ent-16-epi-16-F1t-PhytoP; 9-epi-9-F1t-PhytoP; 9-D1t-PhytoP; 9-epi-9-D1t-PhytoP; 16-B1-PhytoP; and 9-L1-PhytoP; as well as the PhytoFs: ent-16(RS)-9-epi-ST-Δ14-10-PhytoF; ent-9(RS)-12-epi-ST-Δ10-13-PhytoF; and ent-16(RS)-13-epi-ST-Δ14-9-PhytoF were synthesized according to already published procedures [60], and provided by the Institut des Biomolécules Max Mousseron (IBMM) (Montpellier, France). The AQC reagent was purchased from Che-mos GmbH (Regenstauf, Germany). All amino acid standards (histidine (His), 1-methylhistidine (Met-His), 4-hydroxyproline (p-Hyp), asparagine (Asn), phosphoethanolamine (PEA), arginine (Arg), glutamine (Gln), serine (Ser), glycine (Gly), ethanolamine (EA), aspartic acid (Asp), citrulline (Cit), glutamic acid (Glu), threonine (Thr), alanine (Ala), γ-Amino-n-butyric acid (GABA), α-Aminoadipic acid (AADA), proline (Pro), ornithine (Orn), β-Aminoisobutyric acid (BAIB), α-Amino-n-butyric acid (AABA), lysine (Lys), cystine (Cys-cys), cystathionine (Cysta), tyrosine (Tyr), valine (Val), methionine (Met), homocysteine (Hcys-cys), leucine (Leu), isoleucine (Ile), tryptophan (Trp) and phenylalanine (Phe) were obtained from Sigma-Aldrich (Madrid, Spain). Ethanol, water LC-MS quality, dimethyl sulfoxide, ammonium acetate, formic acid and methanol were bought from Panreac (Barcelona, Spain). Acetonitrile was from J.T. Baker (Thermo Fisher Scientific Inc., Waltham, MA, USA). Hexane was obtained from Panreac (Castellar del Valles, Barcelona, Spain); butylated hydroxyanisole (BHA), calcium disodium EDTA and bis–Tris (bis (2-hydroxyethyl) amino-tris (hydroxymethyl) methane) were purchased from Sigma-Aldrich (St. Louis, MO, USA). Boric acid was bought from Probus (Badalona, Spain). The solid-phase extraction (SPE) cartridges used with clean-up of sample purposes were Strata cartridge (Strata X-AW, 100 mg/3 mL), which was acquired from Phenomenex (Torrance, CA, USA).

### 4.6. Qualitative and Quantitative Analysis of Phytohormones

The qualitative and quantitative analysis of phytohormones was carried out according to [6]. Briefly, 0.1 g of fresh leaves from 6 samples per treatment (2 samples per replicate) were crushed in a mortar with liquid nitrogen and stored at −80 °C. Then, they were vortexed with 0.5 mL 80% methanol/water (*v/v*) and incubated at 4 °C for 30 min and finally centrifuged at 15,000 rpm (20,627× *g*), at 4 °C for 15 min. The supernatant was kept in ice and then it was further extracted with 0.5 mL 80% methanol/water (*v/v*) after being incubated and centrifuged under the same conditions described above. Finally, both supernatants from the two previous extractions were passed through Chromafix C18 solid-phase extraction cartridge (Macherey Nagel, Düren, Germany) (previously activated with 3 mL 80% methanol/water (*v/v*). The eluted sample was concentrated to dryness by the use of a rotary vacuum evaporator for approximately 3 h (Speedvac, Thermo, Waltham, MA, USA). Then, the dry residue was resuspended with 200 µL of 20% methanol/water (*v/v*), sonicated for 8 min and filtrated through 0.45 µm polyethersulfone filter (Millipore) and finally injected in an ultra-high-performance liquid chromatography (UHPLC) coupled triple quadrupole mass spectrometry (UHPLC-ESI-QqQ-MS/MS) for qualitative and quantitative analysis [6].

Chromatographic separation of phytohormones and the phytoalexin scopoletin was performed by a method previously described by Gómez-Bellot et al. (2021). Briefly, we used a UHPLC coupled to a 6460 UHPLC-ESI-QqQ-MS/MS (Agilent Technologies, Waldbronn, Germany), using a BEH C18 analytical column (2.1 × 100 mm, 1.7 µm) (Waters, Milford, MA, USA). Mobile phases A (H_2_O) contained 0.01% formic acid (*v/v*) and B acetonitrile. The flow rate was 0.2 mL/min using a linear gradient scheme: (t; %B): (0.0; 19.00), (2.5; 90.00), (4.5; 90.00), (6.00; 19.00), (8.00; 19.00). The injection volume was 10 µL. The column temperatures were 40 °C. The operating conditions for the ionization source were as follows: gas flow: 8 L/min, nebulizer: 45 psi, capillary voltage: 4000 V (positive mode) and 2750 V (negative mode), nozzle voltage: 1000 V (positive mode) and 1500 V (negative mode), gas temperature: 300 °C, sheath gas temperature: 375 °C and jet stream gas flow: 11 L/min. The ion optics and fragmentation conditions are detailed in Table 2. Data acquisition and processing were performed using MassHunter software version B.08.00 (Agilent Technologies). The quantification of the phytohormones and scopoletin detected in the samples was performed according to standard curves freshly prepared each day of analysis using the multiple reaction monitoring mode (MRM) with a predominant transition for quantitative analysis and a second transition for identity confirmation of the compound (MRM transitions detailed in [6]).

### 4.7. Qualitative and Quantitative Analysis of Amino Acids

The extraction method for free amino acids was according to [50,61]. Briefly, 0.1 g of fresh leaves from 6 samples per treatment (2 samples per replicate) were crushed in a mortar with liquid nitrogen and stored at −80 °C. Briefly, 20 mg of powder were homogenized with 500 μL of extraction buffer (MeOH/water, 1:1, *v/v*) using an ultra turrax (IKA, T10, Staufen, Germany), for 30 s on ice. The samples were then incubated on dry ice for 5 min. The homogenates were sonicated in an ultrasound bath for 1 min followed by centrifugation (Eppendorf centrifuge 5804 R, Hamburg, Germany) for 10 min at 17,900× *g*, at 4 °C. The supernatants were transferred to limited volume vials and the precipitates were re-extracted with 500 μL of extraction buffer, homogenized, incubated on ice and centrifuged again. All supernatants were combined. The extracts were immediately derivatized. The derivatization of amino acids and amino thiols was accomplished by following the Waters AccQTagTM Ultra UHPLC amino analysis procedures, as described by [50,61].

Briefly, 350 μL of borate derivatization buffer (0.2 M sodium borate, pH 8.8, with 5 mM calcium disodium EDTA), 50 μL of amino acid standard or jujube extract and 100 μL of re-constituted AQC (10 mM AQC dry powder in acetonitrile) were placed in a 2 mL propylene vial. This solution was vortexed for several seconds, allowed to stand for 1 min at room temperature and then heated in a heating block for 10 min at 55 °C. After removing the vial from the heating block, the sample was injected into a UHPLC-MS/MS. Amino acids and thiols were analyzed by reverse phase UHPLC as reported by [50,62]. Briefly, chromatographic separation was carried out on an Acc Q Tag Ultra BEH column at 20 C (2.1 × 100 mm, 1.7 μm) (Waters Corp., Dublin, Ireland). Two types of eluent were used for gradient separation. Mobile phase A consisted of 50 mL of an aqueous solution (acetonitrile, formic acid and 5 mM ammonium acetate in water) (10: 6: 84, *v/v/v*) diluted with 950 mL of Milli-Q water. Mobile phase B was a mixture of acetonitrile and formic acid (99.9: 0.1, *v/v*). The injection volume was 20 μL and the elution was performed at a flow rate of 0.5 mL/min^–1^. The gradient profile was: 99.9% A at 0–0.5 min, 90.9% A at 5.7 min, 78.8% A at 7.7 min, 40.4% A at 8–10 min, 10% A at 10.01–12.00 min and 99.9% A at 12.01–14.00 min. These compounds were identified using a UHPLC system coupled to a 6460 tandem mass spectrometer (Agilent Technologies, Waldbronn, Germany). Data acquisition and processing were performed using MassHunter software version B.04.00, from Agilent Technologies. The MS analysis was applied in the multiple reaction monitoring (MRM) mode, which was performed using the positive ionization mode. The MS parameters fragmentor (ion optics; capillary exit voltage) and collision energy were optimized for each analyte. The allocation of these parameters, along with preferential MRM transition of the corresponding analytes, generated the most abundant product ions. The MRM transition used for each derivatized amino acid/thiol corresponded, in most cases, to the AMQ moiety (171+), which results from the collision-induced cleavage at the ureide bond of the AMQ adduct of each amino acid/thiol (MRM transition of amino acids detailed in [50]). The working conditions for the MS parameters of the electrospray source were as follows: gas flow, 9 l/min; nebulizer, 40 psi; capillary voltage, 4000 V; nozzle voltage, 1000 V; gas temperature, 325 °C; sheath gas temperature, 390 °C; and jet stream gas flow, 11 l/min. The acquisition time was 12 min for each sample. The most abundant MRM transition of each analyte was used for amino acid quantitation by comparison with its corresponding authentic standard. Calibration standards curves were generated using individual amino acids prepared by dissolving each amine in Bis–Tris (pH 6.5) [50].

### 4.8. Qualitative and Quantitative Analysis of Phytoprostanes (PhytoPs) and Phytofuranes (PhytoFs)

The PhytoPs and PhytoFs present in plant leaves of Melissa officinalis and Mentha spicata were extracted following the methodology described by [63,64] with minor modifications. Briefly, samples (4 g) were pestled with 10 mL of methanolic butylated hydroxyanisole (BHA) (99.9:0.1, *v/w*). The extracts were centrifuged at 2000× *g* for 10 min, and the supernatants were collected and cleaned up by SPE, using Strata X-AW cartridges according to the procedure described [65].

Chromatographic separation of PhytoPs and PhytoFs was performed using a UHPLC coupled with a 6460 triple quadrupole-MS/MS (Agilent Technologies, Waldbronn, Germany), using the analytical column BEH C18 (2.1 mm × 50 mm, 1.7 μm) (Waters, Milford, M.A.). The column temperatures were 6 °C (both left and right). The mobile phases consisted of Milli-Q water/acetic acid (99.99:0.01, *v/v*) (A) and methanol/acetic acid (99.99:0.01, *v/v*) (B). The injection volume and flow rate were 20 μL and 0.2 mL min^–1^ upon the following linear gradient (time (min), %B): (0.00, 60.0%); (2.00, 62.0%); (4.00, 62.5%); (8.00, 65.0%); and (8.01, 60.0%). An additional postrun of 1.5 min was considered for column equilibration. The spectrometric analysis was conducted in multiple reaction monitoring mode (MRM) operated in negative mode, assigning preferential MRM transition for the corresponding analytes. The ionization and fragmentation conditions were as follows: gas temperature 325 °C, gas flow 8 L min^–1^, nebulizer 30 psi, sheath gas temperature 350 °C, jet stream gas flow 12 L min^–1^, capillary voltage 3000 V, and nozzle voltage 1750 V, according to the most abundant product ions. Data acquisition and processing were performed using MassHunter software version B.04.00 (Agilent Technologies). The quantification of PhytoPs and PhytoFs detected in plant samples was performed using authentic standards according to standard curves freshly prepared as mentioned in the previous section. The selected reaction monitoring and chemical names were according to the nomenclature system of [66].

### 4.9. Leaf Mineral Content and Plan Growth

At the 15th day after imposing treatments, the inorganic mineral content of dry leaves was determined in three plants per treatment (one sample per replication) by means of emission spectrophotometry. The leaves were oven dried at 80 °C, ground, and sieved through a 2 mm nylon mesh before analysis. A chemical analysis of water irrigation treatments was performed. The nutrient concentrations were determined in an extract digested with HNO_3_:HClO_4_ (2:1, *v/v*) using an inductively coupled plasma optical emission spectrometer (ICP-OES IRIS INTREPID II XDL).

At the 15th day after imposing treatments, the substrate was gently washed from the roots of three plants per treatment (one per replicate). The plants were divided into leaves and roots. Then, they were oven dried at 80 °C until they reached a constant weight to measure the respective dry weights (DW). Leaf number was estimated and total leaf area (cm^2^) was determined using a leaf area meter (Delta-T; Devices Ltd., Cambridge, UK).

### 4.10. Statistics

In the experiment, all plants were randomly assigned to each treatment, with three replications for each treatment. The data were analyzed by one-way ANOVA using IBM SPSS Statistics 25. Treatment means were separated with Duncan’s multiple range test (*p* ≤ 0.05).

## 5. Conclusions

Despite both species sharing the same family and having a very similar external appearance, the application of saline reclaimed water during a short time period induces a different physiological response. Spearmint plants were less affected than balm plants, showing a certain capacity to tolerate salinity at least up to 1.6 dS m^−1^, while spearmint was also able to cope with saline water up to CE: 4.4 dS m^−1^. The ability of spearmint plants to recover photosynthetic activity from their first negative response was attributed to quick biochemical changes, such as hormonal changes and the accumulation of some amino acids, PhytoP and PhytoF, that induce several plant defence mechanisms. The individual phytofuran *ent*-9(*RS*)-12-*epi*-ST-Δ^10^-13-PhytoF could be postulated as a good marker of the dysregulation of stomatal function in these plants under salt stress conditions. However, despite the decrease in these compounds in the salinity irrigation treatments, they showed a high concentration in both species. Thus, the consumption of these type of plants could provide not only greater organoleptic and nutritional quality but also benefits for human health. It needs to be tested whether a longer irrigation period with saline reclaimed water would change the quality of both species based on biochemical components (hormonal changes and accumulation of amino acids), and based on new types of secondary metabolites (PhytoP and PhytoF) that induce several plant defence mechanisms.

## Figures and Tables

**Figure 1 plants-11-03427-f001:**
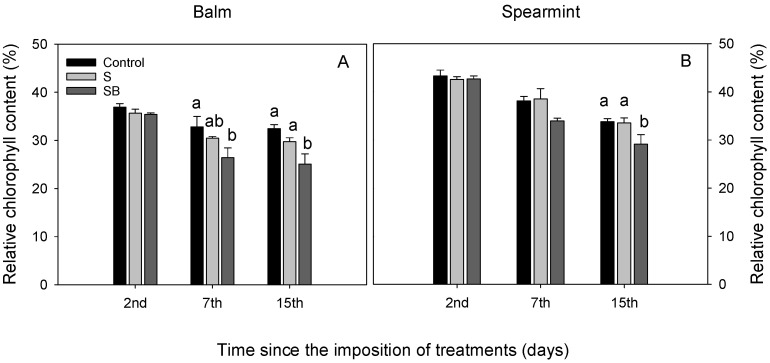
Relative chlorophyll content in balm (**A**) and spearmint (**B**) plants irrigated with water from different sources. Bars without lowercase letters mean no significant differences between treatments. Different lowercase letters indicate significant differences between treatments according to Duncan’s test at *p* ≤ 0.05.

**Figure 2 plants-11-03427-f002:**
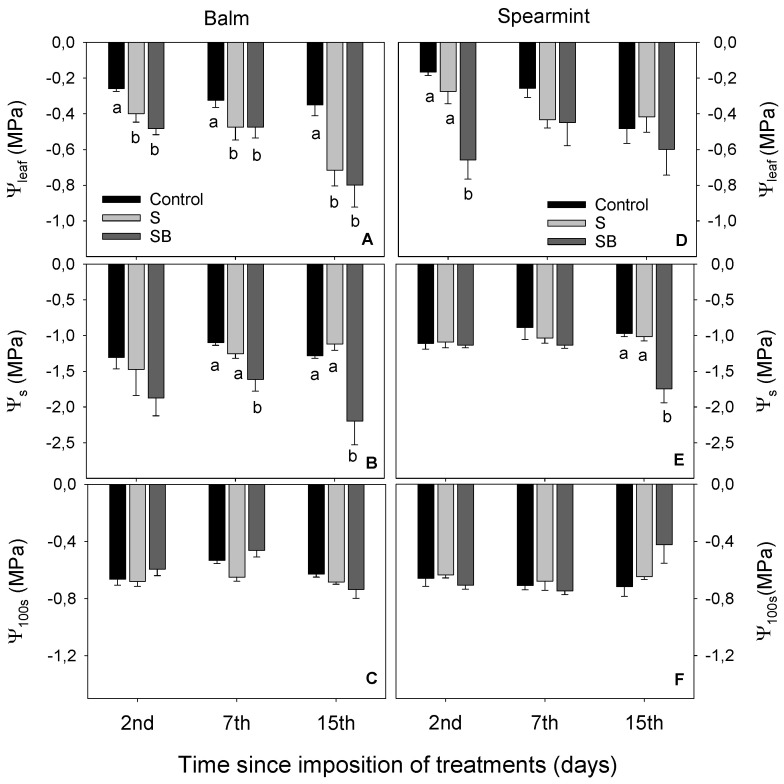
Leaf water potential (Ψleaf) (**A**,**D**), osmotic water potential (Ψos) (**B**,**E**) and osmotic water potential at full turgor (Ψ100s) (**C**,**F**) in balm and spearmint plants irrigated with water from different sources. Bars without lowercase letters mean no significant differences between treatments. Different lowercase letters indicate significant differences between treatments according to Duncan’s test at *p* ≤ 0.05.

**Figure 3 plants-11-03427-f003:**
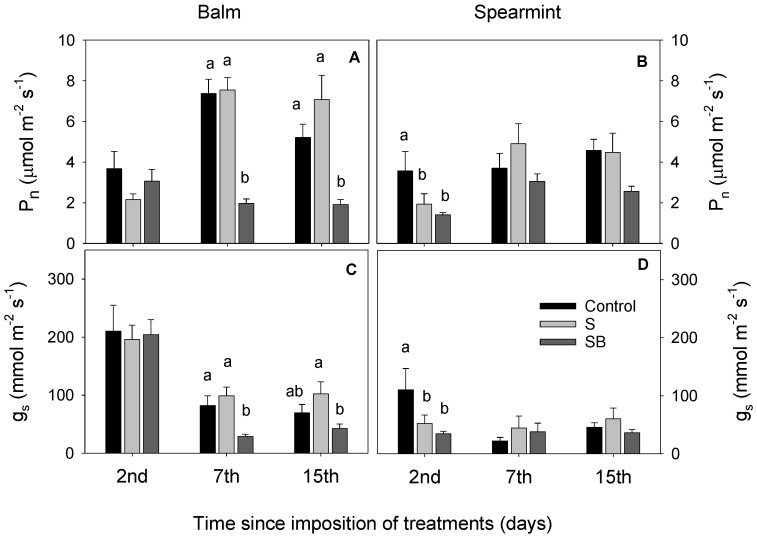
Net photosynthetic rate (Pn) (**A**,**B**) and stomatal conductance (gs) (**C**,**D**) in balm and spearmint plants irrigated with water from different sources. Bars without lowercase letters mean no significant differences between treatments. Different lowercase letters indicate significant differences between treatments according to Duncan’s test at *p* ≤ 0.05.

**Table 1 plants-11-03427-t001:** Leaf mineral content of balm and spearmint plants at 15th day. Values are mean ± S.E. of six plants.

		Mineral Content in Leaves (g 100 g^−1^ of Dry Weight)																	
		Ca^2+^			K^+^			Cl^−^				Na^+^				P			
	**Control**	0.80	±	0.08	4.99	±	0.36	4.72	±	0.24		0.24	±	0.04		0.90	±	0.06	a
**Balm**	**S**	0.70	±	0.17	4.68	±	0.43	4.23	±	0.07		0.20	±	0.10		0.72	±	0.08	ab
	**SB**	0.49	±	0.04	4.58	±	0.16	4.64	±	0.38		0.23	±	0.05		0.62	±	0.02	b
	**P**			ns			ns			ns			ns					*	
	**Control**	1.04	±	0.16	4.40	±	0.59	4.49	±	0.40	b	0.19	±	0.03	b	0.69	±	0.07	a
**Spearmint**	**S**	0.86	±	0.06	3.85	±	0.15	4.42	±	0.20	b	0.18	±	0.02	b	0.57	±	0.03	b
	**SB**	1.12	±	0.17	5.20	±	0.81	6.19	±	0.26	a	0.56	±	0.14	a	0.74	±	0.13	a
	**P**		ns			ns			**					*				*	

Means without a common lowercase letter are significantly different by Duncan 0.05 test. P, probability level; ns, not significant. * *p* ≤ 0.05. ** *p* ≤ 0.01.

**Table 2 plants-11-03427-t002:** Leaf number, leaf area, leaf dry weight (DW) and root dry weight (DW) of balm and spearmint plants at 15th day. Values are mean ± S.E. of six plants.

		Biomass Parameters															
		Leaf Number			Leaf Area (cm^2^)					Leaf DW (g)			Root DW (g)				
	**Control**	386.3	±	27.6	a	178.1	±	6.9	a	11.0	±	0.9	a	9.3	±	1.5	a
**Balm**	**S**	308.0	±	44.0	b	164.2	±	12.1	a	11.2	±	1.0	a	7.0	±	2.0	a
	**SB**	109.0	±	9.0	c	53.5	±	10.3	b	7.5	±	0.4	b	4.3	±	0.2	b
	**P**	**				**				*				*			
	**Control**	392.3	±	77.6		136.5	±	7.3	a	12.5	±	2.12	b	7.7	±	0.8	b
**Spearmint**	**S**	239.3	±	40.2		122.5	±	5.8	a	19.7	±	1.24	a	8.2	±	0.2	b
	**SB**	346.7	±	91.4		51.59	±	16.5	b	12.2	±	1.34	b	10.3	±	1.1	a
	**P**	ns				**				**				*			

Means without a common lowercase letter are significantly different by Duncan 0.05 test. P, probability level; ns, not significant. * *p* ≤ 0.05. ** *p* ≤ 0.01.

**Table 3 plants-11-03427-t003:** Concentration of phytohormones in leaves of balm and spearmint plants irrigated with water with different sources at 15th day. Values are mean ± S.E. of six plants.

		Phytohormones (ng g^−1^)												
		Control			S					SB				P
	**ACC**	18.20	±	2.08	b	15.85	±	1.21	b	180.05	±	17.06	a	***
	**TZ**	33.33	±	8.01	b	21.39	±	2.11	b	64.66	±	11.71	a	**
	**TZ-rib**	0.27	±	0.04	b	0.22	±	0.03	b	0.56	±	0.04	a	***
**Balm**	**TZ-glc**	5.59	±	0.54	a	4.99	±	0.46	a	2.07	±	0.09	b	***
	**ABA**	1.07	±	0.13	a	0.67	±	0.05	b	1.25	±	0.07	a	*
	**SA**	7.25	±	1.52	a	2.17	±	0.26	b	6.88	±	0.45	a	**
	**SC**	33.32	±	4.48		47.28	±	7.99		25.21	±	3.99		ns
	**Spearmint**													
	**ACC**	42.89	±	6.94	b	66.28	±	2.57	a	77.33	±	8.94	a	*
	**TZ**	95.22	±	5.03	a	64.27	±	6.62	b	63.69	±	11.57	b	*
	**TZ-rib**	0.32	±	0.11		0.50	±	0.04		0.69	±	0.15		
**Spearmint**	**TZ-glc**	10.66	±	0.24	a	5.13	±	0.56	b	4.27	±	0.55	b	**
	**ABA**	1.07	±	0.10	b	1.95	±	0.18	a	1.68	±	0.18	a	***
	**SA**	4.17	±	0.64		5.53	±	1.90		5.90	±	1.21		
	**SC**	6.36	±	0.85	b	9.44	±	0.66	a	6.29	±	0.21	b	*

Aminocyclopropane-1-carboxylic acid: ACC. Cytokinins: TZ, TZ-rib, TZ-glc. Abscisic acid: ABA. Salicylic acid: SA. Scopoletin: SC. Different lowercase letters indicate significant differences between treatments according to Duncan’s test at *p* ≤ 0.05. P, probability level; ns, not significant. * *p* ≤ 0.05. ** *p* ≤ 0.01. ****p* < 0.001.

**Table 4 plants-11-03427-t004:** Concentration of amino acids in leaves of balm and spearmint plants irrigated with water from different sources at 15th day. Values are mean ± S.E. of six plants.

		Amino Acids (µg g^−1^)												
		Control			S					SB			P	
	**Ser**	67.7	±	3.8		78.7	±	15.3		103.8	±	31.5		ns
	**Asp**	79.1	±	11.1	b	128.2	±	6.2	a	119.1	±	12.0	a	*
	**Arg**	54.1	±	8.4		50.4	±	9.6		26.9	±	7.0		ns
**Balm**	**P-Hyp**	5.6	±	0.7		5.4	±	0.6		5.7	±	0.7		ns
	**Met-His**	430.1	±	18.3		445.9	±	18.5		437.6	±	18.0		ns
	**Thr**	13.0	±	2.0		13.7	±	0.6		12.6	±	2.9		ns
	**Ala**	16.1	±	1.4		17.4	±	0.9		16.8	±	1.0		ns
	**Leu**	5.5	±	0.6		6.4	±	0.7		6.3	±	0.9		ns
	**Ser**	61.5	±	8.6	b	401.1	±	48.1	b	81.9	±	15.4	a	**
	**Asp**	160.5	±	11.0		200.0	±	35.3		147.9	±	10.3		ns
	**Arg**	46.1	±	4.8		65.5	±	13.7		37.4	±	6.4		ns
**Spearmint**	**P-Hyp**	5.6	±	1.2	b	9.6	±	0.9	a	9.4	±	1.5	a	*
	**Met-His**	413.9	±	8.5	b	433.7	±	12.9	ab	478.4	±	20.5	a	*
	**Thr**	14.4	±	1.9	ab	18.2	±	3.1	a	10.0	±	2.1	b	*
	**Ala**	15.1	±	0.8		16.6	±	0.8		13.2	±	1.6		ns
	**Leu**	5.7	±	0.9	b	10.4	±	0.7	a	12.0	±	1.5	a	**

Serine: Ser. Aspartic acid: Asp. Arginine: Arg. Hydroxyproline: P-Hyp. Methyl-histidine: Met-His. Threonine: Thr. Alanine: Ala. Leucine: Leu. Different lowercase letters indicate significant differences between treatments according to Duncan’s test at *p* ≤ 0.05. P, probability level; ns, not significant. * *p* ≤ 0.05. ** *p* ≤ 0.01.

**Table 5 plants-11-03427-t005:** Concentration of phytoprostanes and phytofurans in leaves of balm and spearmint plants irrigated with water from different sources at 15th day. Values are mean ± S.E. of six plants.

		Phytoprostanes and Phytofurans (µg 100 g^−1^ F.W.)												
Species		Control			S					SB			P	
**Balm**	**9-F_1t_-PhytoP**	0.016	±	0.007		0.014	±	0.002		0.025	±	0.004		ns
	**9-*epi*-9-F_1t_-PhytoP**	0.023	±	0.007		0.025	±	0.007		0.056	±	0.011		ns
	** *ent* ** **-16-*epi*-16-F_1t_-PhytoP + *ent*-16-F_1t_-PhytoP**	0.001	±	0.001		0.002	±	0.002		0.001	±	0.001		ns
	**Total PhytoP**	0.041	±	0.017		0.041	±	0.014		0.077	±	0.012		ns
	** *ent* ** **-9(*RS*)-12-*epi*-ST-** **Δ** ** ^10^ ** **-13-PhytoF**	400.2	±	23,56	a	287.6	±	29,32	b	358.0	±	28,94	ab	*
	** *ent* ** **-16(*RS*)-13-*epi*-ST-** **Δ** ** ^14^ ** **-9-PhytoF**	0.107	±	0.117		0.144	±	0.158		0.144	±	0.157		ns
	** *ent* ** **-16(*RS*)-9-*epi*-ST-** **Δ** ** ^14^ ** **-10-PhytoF**	2.902	±	0.559		2.801	±	0.382		3.188	±	0.261		ns
	**Total PhytoF**	403.29	±	23.81	a	290.04	±	29.57	b	360.76	±	29.13	ab	*
**Spearmint**	**9-F_1t_-PhytoP**	0.043	±	0.009	b	0.087	±	0.015	a	0.018	±	0.006	b	*
	**9-*epi*-9-F_1t_-PhytoP**	0.105	±	0.012		0.199	±	0.039		0.121	±	0.048		ns
	** *ent* ** **-16-*epi*-16-F_1t_-PhytoP + *Ent*-16-F_1t_-PhytoP**	0.001	±	0.001		0.001	±	0.001		0.000	±	0.000		ns
	**Total PhytoP**	0.117	±	0.03	ab	0.247	±	0.075	a	0.096	±	0.045	b	*
	** *ent* ** **-9(*RS*)-12-*epi*-ST-** **Δ** ** ^10^ ** **-13-PhytoF**	299.26	±	17.93	a	273.42	±	19.46	a	191.71	±	24.05	b	**
	** *ent* ** **-16(*RS*)-13-*epi*-ST-** **Δ** ** ^14^ ** **-9-PhytoF**	0.000	±	0.000	b	0.159	±	0.087	a	0.000	±	0.000	b	*
	** *ent* ** **-16(*RS*)-9-*epi*-ST-** **Δ** ** ^14^ ** **-10-PhytoF**	2.286	±	0.327		1.654	±	0.223		1.728	±	0.411		ns
	**Total PhytoF**	301.54	±	19.66	a	274.905	±	19.661	a	193.44	±	24.206	b	**

Different lowercase letters indicate significant differences between treatments according to Duncan’s test at *p* ≤ 0.05. P, probability level; ns, not significant. * *p* ≤ 0.05. ** *p* ≤ 0.01.

**Table 6 plants-11-03427-t006:** Physicochemical analysis of irrigation water from different sources (control, C; secondary effluent, S; and secondary effluent with brine, SB). Data are values from samples collected at the beginning of the experiment.

Physicochemical Analysis	C	S	SB
pH	8.6	8.52	8.43
CE (dS m^−1^)	1.14	1.55	4.37
B (mg L^−1^)	0.06	0.12	0.22
Ca (mg L^−1^)	39.26	48.84	127.90
Fe (mg L^−1^)	<0.01	0.09	0.01
K (mg L^−1^)	9.57	29.25	75.15
Mg (mg L^−1^)	39.35	34.45	87.57
Mn (mg L^−1^)	0.36	0.32	0.68
Na (mg L^−1^)	172.00	219.20	828.60
P (mg L^−1^)	<0.01	1.58	12.09
S (mg L^−1^)	95.90	79.58	338.90
Zn (mg L^−1^)	0.35	0.04	0.05
Ni (mg L^−1^)	<0.01	<0.01	0.01
Cl^−^ (mg L^−1^)	214.62	317.87	1066.08
NO_2_^−^ (mg L^−1^)	<0.1	<0.1	<0.1
Br^−^ (mg L^−1^)	0.21	0.33	32.22
NO_3_^−^ (mg L^−1^)	13.08	26.87	41.29
PO_4_^3−^ (mg L^−1^)	<1.0	<1.0	<1.0
SO_4_^2−^ (mg L^−1^)	338.33	273.18	1067.69

## Data Availability

Not applicable.

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
