# Peer review of "Acute and Rapid Response of Melissa officinalis and Mentha spicata to Saline Reclaimed Water in Terms of Water Relations, Hormones, Amino Acids and Plant Oxylipins"

_plants, 2022, doi:10.3390/plants11243427_

Round 1

Reviewer 1 Report

The authors investigated the "Acute and rapid response of Melissa officinalis and Mentha spicata to saline reclaimed water in terms of water relations, hormones, amino acids, and plant oxylipins". This is an exciting area of research. The authors collected a large amount of data. The presentation was impressive. Before publishing this manuscript I would like to suggest the following

1.  Use a similar font both in TEXT and Table/Figure

2. The Bars without lowercase letters are difficult to understand. please Explain

3. Use appropriate and similar legends for all graphs and tables i.e., C, S, and SB. Also give the elaboration of C, S, and SB in all figures and tables

4. Did you determine Ca, K, Cl, Na OR Ca2+, K+, Cl-, Na+? Please double-check it.

5. What does ± means in the table? Please explain.

Reviewer 2 Report

The present manuscript entitled “Acute and Rapid Response of Melissa Officinalis and Mentha Spicata to Saline Reclaimed Water in terms of Water Relations, 3 Hormones, Amino Acids and Plant Oxylipins submitted by Gómez-Bellot et al., reveals that both species (spearmint and Balm) belong to the same family but their responses are different against saline reclaimed water. Where spearmint was more tolerant to salinity than plam because spearmint  showed better physiological efficiency.  The outcomes of this study are quite clearly presented in the abstract. The paper's theme and presentation are appropriate and helpful to readers. The MS is in good shape and mostly well written other than where I've specifically noted section of MS that could be improved.

Why did authors use one-way ANOVA and two- way ANOVA?

Authors are suggested to rewrite this sentence “Despite both species share the same family and have very similar external appear- 551 ance, the application of saline reclaimed water during a short time period induces that 552 spearmint were less affected than balm plants, who showed a certain capacity to tolerate 553 salinity at least up to 1.6 dS m-1, while spearmint was able to cope with saline water up to 554 CE: 4.4 dS m-1.” 

Reviewer 3 Report

General remark:

What's the point of the "S" variant  if the chemical composition does not show significant differences from the control solution? I'm not sure that it is close to salinity, on the contrary, the solution «S» is richer in iron, potassium, phosphorus, nitrogen. Therefore, the observed differences in plants of the “S” variant from control plants are the norm of the reaction under slightly changed conditions.

 Introduction

Predominantly well written, logical and understandable.

Lines 73-75 «But it is not clear which mechanisms plants employ to maintain residual growth and to what extent these mechanisms differ between short- and long-term responses» – old reference [14]

Regarding goals. Spicy and aromatic plants were studied. Why was the goal not to analyze aromatic oils, the most valuable substances from these plants, but products of lipid peroxidation?

Methods

Line 473 "ace-tonitrile" is an extra dash.

Line 536 - Specify temperature and duration of digestion

Results

Photos of plants are highly desirable, 2-3 pieces per variant.

Fig 2 - Days 1, 2 and 3 are indicated on the time axis. The results say about the 15th day. Caption error?

Lines 146-148  - It is not clear why this piece of text? What is CRC?

Table 1 - What weight was calculated? Must be converted to dry weight.

Table 3 and 4 - mistake in the name of the variant "SB"?

Table 5 - Is not a good abbreviation for the species name. There is confusion with other designations.
